



# Characterization of NR-PM₁ and source apportionment of organic aerosol in Krakow, Poland

Anna K. Tobler[1,2], Alicja Skiba[3], Francesco Canonaco[2], Griša Močnik[4,5], Pragati Rai[1], Gang Chen[1], Jakub Bartyzel[3], Miroslaw Zimnoch[3], Katarzyna Styszko[6], Jaroslaw Nęcki[3], Markus Furger[1], Kazimierz Różański[3], Urs Baltensperger[1], Jay G. Slowik[1] and André S. H. Prévôt[1]

[1]Laboratory of Atmospheric Chemistry, Paul Scherrer Institute, 5232 Villigen PSI, Switzerland
[2]Datalystica Ltd., Park innovAARE, 5234 Villigen, Switzerland
[3]AGH University of Science and Technology, Faculty of Physics and Applied Computer Science, Department of Applied Nuclear Physics, Krakow, Poland
[4]Condensed Matter Physics Department, J. Stefan Institute, Ljubljana, Slovenia
[5]Center for Atmospheric Research, University of Nova Gorica, Ajdovščina, Slovenia
[6]AGH University of Science and Technology, Faculty of Energy and Fuels, Department of Coal Chemistry and Environmental Sciences, Krakow, Poland

*Correspondence to*: André S.H. Prévôt (andre.prevot@psi.ch)

**Abstract.** Krakow is routinely affected by very high air pollution levels, especially during the winter months. Although a lot of effort has been done on characterization of ambient aerosols, there is a lack of online and long-term measurements of non-refractory aerosols. Our measurements at AGH University provide online long-term chemical composition of ambient submicron particulate matter (PM₁) between January 2018 and April 2019. Here we report the chemical characterization of non-refractory submicron aerosols and source apportionment of the organic fraction by positive matrix factorization (PMF). In contrast to other long-term source apportionment studies, we let a small PMF window roll over the dataset instead of performing PMF over the full dataset or on separate seasons. In this way, the seasonal variation of the source profiles can be captured. The uncertainties of the PMF solutions are addressed by the bootstrap resampling strategy and the random *a*-value approach for constrained factors.

We observe clear seasonal patterns in concentration and composition of PM₁, with high concentrations during the winter months and lower concentrations during the summer months. Organics are the dominant species throughout the campaign. Five organic aerosol (OA) factors are resolved, of which three are of primary nature (hydrocarbon-like OA (HOA), biomass burning OA (BBOA) and coal combustion OA (CCOA)) and two are of secondary nature (more oxidized oxygenated OA (MO-OOA) and less oxidized oxygenated OA (LO-OOA)). While HOA contributes on average 8.6 % ± 2.3 % throughout the campaign, the solid fuel combustion related BBOA and CCOA show a clear seasonal trend with average contributions of 10.4 % ± 2.7 % and 14.1 %, ± 2.1 % respectively. The highest contributions are observed during wintertime as a result of residential heating. Throughout the campaign, the OOA can be separated into MO-OOA and LO-OOA with average contribution of 38.4 % ± 8.4 % and 28.5 % ± 11.2 %, respectively.



# 1    Introduction

Aerosols adversely impact visibility, air quality (Watson, 2002), public health (Pope and Dockery, 2006) and regional to global climate (IPCC, 2013). A significant fraction of the aerosol mass consists of organic aerosol (OA) particles (Jimenez et al., 2009). The knowledge of OA properties and the characterization of the OA chemical composition and different sources has greatly improved over the last decades. The aerosol chemical speciation monitor (ACSM, (Ng et al., 2011b)) is widely used by many research groups and air quality monitoring agencies to quantify the chemical composition of non-refractory (NR)

particulate matter (PM), because it allows for online long-term measurements of NR-PM with little maintenance needed. Online techniques have the advantage to provide (near) real-time measurements of the aerosol composition and properties while minimizing the risks of sample contamination and losses during the sample preparation. A good inter-comparability of different ACSMs has been shown in previous studies (Crenn et al., 2015; Freney et al., 2019).

Cities in Eastern Europe experience poor air quality from both a European and global perspective (European Environment

Agency, 2019). Despite high levels of ambient pollution, online state-of-the-art atmospheric research of the chemical composition of the ambient aerosol remains scarce. There are only few studies from Eastern Europe, including the Czech Republic (Kubelová et al., 2015), Romania (Vasilescu et al., 2017) and Estonia (Elser et al., 2016a). Especially in Southern Poland, the annual limit of 25 $\mu g \cdot m^{-3}$ regarding particulate matter with an aerodynamic diameter smaller than 2.5 µm (PM$_{2.5}$) is substantially exceeded in many places (European Environment Agency, 2019), as it is also the case for Krakow, where long-

term online measurements of the chemical characterization of ambient aerosol are not available yet.

Previous air quality studies in Krakow were mostly based on offline analysis, focused on elemental analysis (Samek et al., 2017; Samek et al., 2019; Zimnoch et al., 2020) or oxidative potential (Styszko et al., 2017). These studies identified coal, traffic, steel, cement and metal industries as major pollution sources. In Poland, the combustion of coal is very common. For example, in 2016 over 80 % of the electricity in the country is produced through coal combustion (International Energy

Agency, 2017). Also residential heating strongly depends on coal combustion, as it is estimated that almost 70 % of the houses have a coal-fired boiler installed (Zaborowski and Dworakowska, 2016). Coal is a heavily polluting fuel, especially during incomplete combustion. An earlier study of real-time PM$_{10}$ measurements identified residential heating by coal combustion as the major contributor to severe smog episodes in Krakow (Mira-Salama et al., 2008). Krakow passed a resolution to ban the use of all solid fuels within the city, starting from 1 September 2019 (Marshal's Office of the Malopolska Region, 2016). A

thorough understanding of the chemical composition of PM is crucial for the evaluation of this policy's efficiency, as well as in the design of future actions.

In this study, we present a detailed characterization of NR-PM$_1$ based on the first long-term ACSM measurements in Poland. We identified and quantified five main OA sources based on positive matrix factorization analysis (PMF, (Paatero and Tapper, 1994)). The advanced rolling window technique was applied during the PMF analysis (Canonaco et al., 2021) which allows

us to capture the temporal variability of the emission sources related to systematic atmospheric variability over the





measurement campaign and in addition allows estimating the uncertainties of the PMF analysis. This work contributes to a better understanding of ambient aerosols and smog episodes in Krakow before the ban of solid fuels.

## 2    Methodology

### 2.1    Sampling site and instrumentation

The long-term submicron aerosol online measurements presented in this study were performed from 8 January 2018 until 10 April 2019 at the Faculty of Physics and Applied Computer Science, AGH University of Science and Technology (50°04' N, 19°55' E) in Krakow, Poland. The faculty building is located in the Krowodrza district, west of the old city center of Krakow, in a suburban residential area with high traffic intensity, recreational areas and residential buildings.

A quadrupole aerosol chemical speciation monitor (Q-ACSM, Aerodyne Research Inc., Billerica, MA, USA) (Ng et al., 2011b)
and a seven-wavelength aethalometer model AE33 (Magee Scientific, Berkeley, CA, USA) (Drinovec et al., 2015) were installed in a temperature-controlled room. Ambient aerosols were continuously sampled at a flow rate of 5 L min$^{-1}$ through a PM$_{2.5}$ cyclone (BGI, Mesa Labs, Inc.), installed 2 m above the rooftop of the building. The measurements were conducted in the room below. A by-pass flow of 2 L min$^{-1}$ was installed to maintain the total flow required for the cyclone. Aerosols were dried by a Nafion dryer (MD-110-24S-4, Perma Pure LLC). Afterwards, the flow was split so that the ACSM sampled
approximately 0.1 L min$^{-1}$, while a 3 L min$^{-1}$ by-pass flow was maintained to ensure near-isokinetic sampling conditions for the ACSM. From 8 January 2018 to 22 June 2018, the aethalometer was installed in front of the Nafion dryer with a sample flow of 2 L min$^{-1}$, afterwards the aethalometer was used to uphold the by-pass flow at the ACSM and sampled with a flow rate of 3 L min$^{-1}$.

### 2.2    Quadrupole aerosol chemical speciation monitor (Q-ACSM)

The Q-ACSM sampling method and operating details have been previously described in detail by Ng et al. (2011b). Briefly, the aerosols are focused in an aerodynamic lens (PM$_1$, (Liu et al., 2007)) into a narrow beam after transmission through a 100 μm critical orifice. After passing through a differentially pumped vacuum chamber, non-refractory particles are flash-vaporized on a tungsten vaporizer operated at ~600 °C. An yttriated iridium filament is then used to ionize the evolved vapors via electron impact at 70 eV. The ions are subsequently detected by a quadrupole residual gas analyzer (RGA, Pfeiffer Vacuum
Prisma Plus). The ACSM alternately samples ambient air and filtered, particle-free air, each for 30 s. The difference spectrum of these measurements represents the aerosol mass spectrum, which typically ranges between mass to charge ratios *m/z* 10 and 150 with unit mass resolution (UMR).

To quantify the mass concentration, the Q-ACSM was routinely calibrated with ammonium nitrate (NH$_4$NO$_3$), ammonium sulfate ((NH$_4$)$_2$SO$_4$), and ammonium chloride (NH$_4$Cl) in order to determine the response factor (RF) of the instrument and
the relative ionization efficiencies (RIE) of ammonium, sulfate, and chloride. Because of artifacts related to slowly vaporizing



species, the instrument showed apparent negative chloride concentrations for most of the campaign. This apparent negative chloride concentration was corrected by adaptation of the standard fragmentation table in combination with an instrument specific RIE_Chl' based on measurement of $m/z$ 36 (HCl$^+$), as described by Tobler et al. (2020b). For this study, an average RIE_Chl' = 0.41 ± 0.17, $RIE_{NH4}$ = 2.43 ± 0.58 and $RIE_{SO4}$ = 0.38 ± 0.11 together with an average $RF_{NO3}$ = 4.68 ± 1.66 · 10$^{-11}$

amps (μg m$^{-3}$)$^{-1}$ were employed. The particle collection efficiency (CE) was assessed using the method Middlebrook et al. (2012). It resulted in a constant CE of 0.5, as the particles were dried, no sufficiently acidic particles were measured and the ammonium nitrate mass fraction (ANMF) was below 0.4 for > 99.7 % of the data. Based on the data acquired during the calibrations, an inorganic salt interference between 3.6 % and 7.8 % on $m/z$ 44 was found following the calculations of Pieber et al. (2016). Although the instrument was calibrated regularly, the full extent of the artefact was hard to quantify, so this

correction could also introduce additional uncertainties to the dataset. Furthermore, it has been shown that discrepancies in the $f_{44}$ (fraction of $m/z$ 44 to organic mass) can result in significant differences in the PMF factor profile analysis but not in the total factor contribution (Fröhlich et al., 2015). Therefore, the data presented here does not include the suggested change in the fragmentation table for $m/z$ 44.

The ACSM data was analyzed using ACSM Local 1.6.1.3 (Aerodyne Research Inc.) in Igor 6.37 (Wavemetrics Inc.). The data

was collected with a resolution of 10 min and then averaged to 30 min for the PMF analysis. The time is reported in UTC (1 and 2 hours behind local time during winter and summer, respectively).

### 2.2.1    Aethalometer

The Magee Scientific Aethalometer model AE33 measures the light attenuation at seven wavelengths (370, 470, 520, 590, 660, 880 and 950 nm) of particles collected on a filter tape (M8020; described in Drinovec et al. (2015), and M8060). The

light attenuation is converted into equivalent black carbon (eBC) mass concentrations using the nominal mass absorption cross section (MAC) value of 7.77 m$^2$ g$^{-1}$ (at 880 nm).  The MAC value used here is used in conjunction with the multiple scattering parameter C appropriate for the tape (Drinovec et al., 2015; Yus-Díez et al., 2021) as discussed in the Supplement (Sect. S1). The dual-spot technique of the AE33 is able to correct loading non-linearities in real-time (Drinovec et al., 2015; Drinovec et al., 2017). This correction was checked using BC(ATN) plots (Drinovec et al., 2015) for the wavelengths used in source

apportionment (see below and Fig. S1).

Different combustion sources feature different light absorption wavelength dependence, especially in the ultraviolet (UV) and lower visible range, a feature that can be used for source apportionment (Sandradewi et al., 2008; Zotter et al., 2017). Sandradewi et al. (2008) presented a model to separate biomass burning (eBC$_{wb}$) and traffic (eBC$_{tr}$) contributions in environments dominated by those two combustion types.

A single parameter is used to describe the source specific dependence of the light absorption coefficient $b$ on the wavelength – the absorption Ångström exponent (AAE):





$$AAE = \frac{\ln\left(\frac{b_{470\,nm}}{b_{950\,nm}}\right)}{\ln\left(\frac{950}{470}\right)} \tag{1}$$

Zotter et al. (2017) proposed Ångström exponents of 0.9 and 1.68 for traffic and wood burning based on $^{14}$C elemental carbon (EC) measurements, respectively. If no such measurements are available, the absorption AAE frequency distribution allows
an estimation of the AAE values that should be used in the model. We expect to separate traffic emissions from solid fuel (biomass burning and coal combustion) emissions. Based on the AAE probability density function (Fig. S2) source specific values for traffic $AAE_{tr} = 0.85$ and solid fuel $AAE_{sf} = 1.9$ were used for the eBC source apportionment.

### 2.2.2    Additional measurements

Meteorological parameters were measured on the rooftop by conventional methods. From 1 January 2019 on, concentrations
of the trace gases CO (Horiba APMA-360CE CO Analyzer), $O_3$ (Thermo Scientific 49i Ozone Analyzer), $NO_x$ (Horiba APMA-360CE $NO_x$ Analyzer) and $SO_2$ (Thermo Scientific 43i $SO_2$ Analyzer) were monitored. Since the gas measurements at AGH are not available for the full campaign, reference data from the monitoring station ("Bujaka station") run by the Chief Inspectorate for Environmental Protection, located ca. 6.8 km south-east of AGH monitoring station (50°00' N, 19°57' E)was used as well. The Bujaka station routinely measures $NO_x$ (Teledyne API T200 NO/NO2/NOx Analyzer), $O_3$ (Environnement
S.A. Model O342e UV Photometric Ozone Analyzer), $SO_2$ (Teledyne API T100 UV Fluorescent $SO_2$ Analyzer) as well as $PM_{2.5}$ (BAM-1020 Met One Instruments).

From 15 March to 10 April 2019, an Xact 625i® Ambient Metals Monitor (Cooper Environmental, Beaverton OR, USA) was installed on the rooftop next to the ACSM inlet. The Xact determines the elemental concentrations in ambient aerosols by X-ray fluorescence (Furger et al., 2017). It was set up with an automated alternating $PM_{2.5}$ and $PM_{10}$ inlet (Furger et al., 2020) to
quantify 32 elements (Al, Si, P, S, Cl, K, Ca, Sc, Ti, V, Cr, Mn, Fe, Co, Ni, Cu, Zn, Ga, Ge, As, Se, Br, Rb, Sr, Y, Zr, Cd, In, Sn, Sb, Ba and Pb) with 1 h time resolution (Rai et al., 2021). In this work, only the $PM_{2.5}$ measurements are used as it is better comparable to the ACSM data ($PM_1$).

### 2.3    OA Source apportionment

Positive matrix factorization (PMF) (Paatero and Tapper, 1994) is a bilinear receptor model with non-negativity constraints,
which describes the variability of a multivariate dataset. Here, it is applied to the organic mass spectra measured by the Q-ACSM. The dataset **X**, i.e. the time series of organic mass spectra, is represented by the matrix product of factor contributions **G** and factor profiles **F**. The fraction that cannot be explained by the model is contained in the residual matrix **E**.

$$\mathbf{X = GF + E} \tag{2}$$



While **G** describes the time series (mass concentration) of a certain source, **F** represents the chemical fingerprint (mass
spectrum) of the source. The dimensions of **G** and **F** depend on the rank $p$, the number of factors chosen to describe the data
set. PMF minimizes the quantity $Q$ (Eq. (3)), defined by the elements of the residual matrix **E** ($e_{ij}$) and the measurement
uncertainty ($\sigma_{ij}$) by using a least squares algorithm in order to solve Eq. (2):

$$Q = \sum_{i=1}^{n} \sum_{j=1}^{m} \left(\frac{e_{ij}}{\sigma_{ij}}\right)^2 \tag{3}$$

PMF suffers from rotational ambiguity, where different combinations of **G** and **F** lead to similar $Q$ values. Some of those
solutions may be mixed and/or not be environmentally reasonable. It has been shown that introducing constraints based on *a
priori* information is an effective method to separate environmentally reasonable solutions (Canonaco et al., 2013; Crippa et
al., 2014). The multilinear engine (ME-2) algorithm (Paatero, 1999) allows the introduction of factor profiles and time series
constraints in form of the *a*-value approach. In case of a profile constraint, the *a*-value defines the extent a factor is allowed to
vary from the anchor profile during the PMF iteration:

$$f_{j,solution} = f_j \pm a \cdot f_j \tag{4}$$

The statistical uncertainty (stability) of the PMF solution can be evaluated using the bootstrap resampling strategy (Davison
and Hinkley, 1997). During each iteration of the bootstrap analysis, the entries of the input data matrix and corresponding
entries of the error matrix are randomly resampled. The newly created matrices have the same size as the original input
matrices.

### 2.3.1    Rolling PMF technique

A disadvantage of PMF for analysis of long-term (i.e., multi-season) datasets is the assumption of static factor profiles over
the entire period PMF performed. While this may be a reasonable approximation for short-term measurements, long-term
measurements, as typical for ACSM, are likely subject to evolving factor profiles due to seasonality. To account for such
source variations, the rolling PMF window technique has been introduced (Canonaco et al., 2021; Parworth et al., 2015). In
this approach, PMF is performed over a small window, which is gradually translated across the entire dataset. The user selects
the width of the PMF window, the shift parameter and the PMF repeats per window based on the dataset. The settings used in
analysis of this dataset are presented in Sect. 3.2.2. This approach generates an enormous amount of PMF solutions, which
requires automated criteria to identify and accept environmentally reasonable PMF solutions (while rejecting the others). This
is done based on seasonal pre-tests and is further discussed in Sect. 3.2.1. Chen et al. (2020) demonstrated that the rolling PMF
approach provides better solutions regarding scaled residuals and comparison with external data compared to traditional
seasonal PMF. The presented OA source apportionment was conducted through the Source Finder Professional (SoFi Pro,





Datalystica Ltd., Villigen, Switzerland) (Canonaco et al., 2013; Canonaco et al., 2021) within the Igor Pro software environment (Wavemetrics, Inc., Lake Oswego, OR, USA).

## 3 Results and discussion

### 3.1 Chemical composition and seasonal variations of NR-PM$_1$

In this study, we report the first long-term online measurements of non-refractory (NR) particulate matter with a diameter < 1 µm (PM$_1$) for Krakow, Poland. The measurements were carried out from 8 January 2018 until 10 April 2019. The time series of the chemical composition of NR PM$_1$ species (Chl, NH$_4$, NO$_3$, Org and SO$_4$) are shown in Fig. 1. A clear seasonal pattern was observed with highest concentrations during the winter months and lowest concentrations during summer. Monthly average concentrations ranged from 15.9 µg·m$^3$ (May 2018) to 61.1 µg·m$^3$ (January 2019). Hourly peak concentrations reached up to 251.2 µg·m$^3$ (January 2018).

Based on monthly averages, the fraction of sulfate was stable during the year (between 13.1 % and 19.7 %), while chloride presented the strongest seasonal variation. In January 2018, the fraction of chloride was 18 %, whereas in June 2019 chloride only contributed 0.8 % to the total NR-PM$_1$ mass. This variation is associated with enhanced coal combustion emissions in winter and the gas-particle phase equilibrium shift due to elevated temperatures in summer, meaning that during warmer weather chloride is mostly present in the gas phase as HCl. Organic aerosol (OA) was the dominant species through the entire campaign, with its contribution to the total mass ranging from 38 % in February 2018 to 55 % in September 2018. The seasonal variation and source contributions to the total OA mass is discussed in more detail in Sect. 3.2.

The diurnal variations of the ACSM species over all seasons are shown in Fig. S3. The diurnal cycle of the organics was similar over all the seasons. The cycle showed the highest concentrations overnight, a small shoulder in the early morning and the lowest concentrations during early afternoon. PMF analysis (Sect. 3.2) revealed that the evening peak was driven by primary emissions. Nitrate showed a pronounced peak in the morning hours for all seasons. Ammonium showed a similar pattern but with more stable diurnal profile in summer and fall. The diurnal pattern of chloride was most dominant during the colder seasons and was characterized by high concentrations during the night. Coal combustion was found to co-emit high chloride levels (Iapalucci et al., 1969; Yudovich and Ketris, 2006). The chloride diurnal cycle was likely driven by coal combustion emissions from residential heating and temperature variations, through temperature dependent gas-particle partitioning. Sulfate showed the most stable diurnal cycle of all species. The elevated concentrations in the afternoon in the warmer seasons are likely due to photochemical processing of gas-phase SO$_2$.

### 3.2 OA source apportionment

PMF was performed on the organic aerosol fraction, averaged to 30 min resolution to improve the signal to noise ratio. Since the presence of coal combustion OA (CCOA) was expected, which has a typical marker signal at *m/z* 115, the mass-to-charge



range of *m/z* 12 to 120 was considered for PMF analysis. However, *m/z* 12 and *m/z* 37 were excluded because of systematically negative signals during most of the campaign.

### 3.2.1     Pretests and definition of environmentally reasonable PMF solutions

PMF analysis of individual seasons was performed to identify the major sources and to define the criteria for environmentally reasonable PMF solutions in the rolling analysis. The dataset was divided into the four seasons (winter (December, January, February), spring (March, April, May), summer (June, July, August) and fall (September, October, November)). Since the campaign covers a period of 16 months, which includes two winters and two springs, six separate PMF inputs were prepared.

In a first step, the primary factors were separated. Therefore, the winter season was thoroughly investigated to capture and
separate the solid fuel sources. This season was chosen as it was expected to exhibit the highest emissions of biomass burning OA (BBOA) and coal combustion OA (CCOA) due to residential heating. Unconstrained PMF did not lead to clean primary factor profiles. To prevent the mixing of the hydrocarbon-like OA (HOA) and the solid fuel OA, the HOA profile was constrained by the reference profile by Crippa et al. (2013) from a short-term AMS study in Paris. This reference profile has proven to be very stable even if there are differences in the vehicle fleet (Tobler et al., 2020a). There was no evidence of a
cooking OA (COA) factor. The high fractions of *m/z* 60 and *m/z* 73 suggested the presence of BBOA. Therefore, BBOA was constrained with different *a*-values by using the reference profile by Ng et al. (2011a). Analysis of the *a*-value sensitivity tests gave a slightly altered BBOA factor profile, adapted to the current dataset. Using this newly found BBOA over the initial BBOA reference profile for further analysis was beneficial as the residual could be decreased. Thanks to the constraining of HOA and BBOA, a third primary factor could be resolved, which would have most likely remained mixed in a fully
unconstrained PMF analysis. This factor exhibited signals from unsaturated hydrocarbons and polycyclic aromatic hydrocarbons (PAHs) as typically found in CCOA profiles. Especially hydrocarbons in the higher mass range (*m/z* 77, 91, 105 and 115) were mostly explained by this primary factor (explained variation by up to 0.82, 0.76, 0.81 and 0.91, respectively). Since the coal profiles may strongly depend on the type of coal, the exploration of this factor profile was preferred over the CCOA reference profiles from coal used in China (Elser et al., 2016b) or Ireland (Dall'Osto et al., 2013; Lin et al., 2017). To
test the stability of the CCOA factor found, the input data matrix was repeatedly ($n = 250$) perturbed using the bootstrap resampling strategy (Davison and Hinkley, 1997). The obtained CCOA factor profile was then used to constrain the solution together with HOA and BBOA. The number of oxygenated OA (OOA) factors was decided based on residual analysis. Although, it is more common to have less separation of the OOA sources in winter compared to summer, the high residuals at *m/z* 60 and *m/z* 73 could only be decreased by allowing two OOA factors being present in all seasons. Increasing the number
of factors to 6 or more factors led to splitting of either the OOAs or the CCOA. Therefore, the 5-factor solution (with two OOAs) was favored throughout the campaign.

Based on these seasonal pretests, PMF with the rolling window approach was carried out. The exact settings applied in this study are described in Sect. 3.2.2. As this approach led to an enormous amount of single PMF solutions, carefully chosen user-





defined criteria were needed to define and separate environmentally reasonable PMF solutions. Solutions that fulfilled all
criteria simultaneously were regarded as environmentally reasonable solutions. These criteria were also used to identify and
sort the unconstrained factors. In the following, specific characteristics of the factors are discussed.

$eBC_{tr}$ is a common tracer for traffic emissions, therefore it is used as the criterion for HOA. Although the aethalometer model
(Sandradewi et al., 2008) was developed for environments dominated only by two combustion sources, namely traffic and
biomass burning, the model still works well enough to separate liquid (traffic) and solid fuels (biomass burning and coal
combustion). Moreover, it can be assumed that the AAE for traffic is relatively constant throughout the year and therefore the
$eBC_{tr}$ is trustworthy. Correlations of $eBC_{tr}$ with HOA were subject to a student $t$-test to evaluate whether the difference in the
correlation was significant compared to the correlation of $eBC_{tr}$ with other factors. Solutions with a $p$-value $\leq 0.01$ were
accepted as reasonable.

Anhydrous sugar fragments, such as levoglucosan fragments ($m/z$ 60 and $m/z$ 73) are typical for BBOA. Therefore, the
explained variation of $m/z$ 60 was chosen as a criterion. Levoglucosan can also be found in coal combustion emissions (Fabbri
et al., 2008; Elser et al., 2016b), but comprises a much smaller fraction than in wood burning. Hence, only solutions that had
higher explained variation of $m/z$ 60 in BBOA compared to CCOA or the OOAs were accepted as environmentally reasonable.

CCOA is characterized by signals from unsaturated hydrocarbons and polycyclic aromatic hydrocarbons (PAHs) at $m/z$ 41,
51, 53, 55, 69, 77, 91 and 115 (Dall'Osto et al., 2013; Elser et al., 2016b; Lin et al., 2017; Xu et al., 2020). The explained
variation of $m/z$ 115 was chosen as a criterion, as it exhibited the highest explained variation among those ions (in winter up
to 0.91) and had less potential sources compared to e.g. $m/z$ 55 or $m/z$ 69. Similar to the explained variation of $m/z$ 60 in BBOA,
solutions with the highest explained variation of $m/z$ 115 in CCOA than in other factors were regarded as environmentally
reasonable. In addition, coal combustion is often accompanied by high chloride emissions (Iapalucci et al., 1969; Yudovich
and Ketris, 2006). In the winter solutions, the correlation of CCOA and Chl was exceptionally high. However, in the summer
months, the correlation may be quite weak, as the gas-particle phase equilibrium changes drastically with increased
temperatures. While chloride is mostly present in the particle phase as $NH_4Cl$ when lower temperatures are present, with higher
temperatures chloride is mostly present in the gas phase as HCl. Under such conditions, Chl levels may be close to the detection
limit of the ACSM and do not provide a reliable reference for CCOA concentrations. Therefore the correlation of CCOA and
Chl was only used as a criterion during the period where the daily Chl averages were consistently well above the detection
limit, meaning that this criterion was not evaluated from 31 March to 2 November 2018.

The two unconstrained OOA factors were separated into more oxidized oxygenated OA (MO-OOA) and less oxidized
oxygenated OA (LO-OOA) based on the fraction of $m/z$ 44 in the factor profiles. This sorting criterion proved more robust
compared to sorting based on the ratio of $m/z$ 44 to $m/z$ 43 in the profiles of the OOA factors. Typically, the ambient OOAs
are represented in the $f_{44}/f_{43}$ space (Ng et al., 2010). To avoid unreasonable solutions with zero intensities at those two $m/z$'s,
solutions with $f_{43} \leq 0.01$ and/or $f_{44} \leq 0.01$ were rejected.


### 3.2.2    Rolling PMF settings

After evaluation of the seasonal PMF solutions, rolling PMF was performed. The rolling PMF approach is defined by the shift parameter (amount of days by which the PMF window is shifted), the width of the PMF window (amount of consecutive days over which PMF is performed) and the number of repeats per PMF window. The PMF window was always shifted by 1 day
in this study to capture variations of the emission sources best (Canonaco et al., 2021). For this study window lengths of 7, 14, 21 and 28 days were tested. The same set of criteria and thresholds was used on all four different PMF analyses to better compare them. Previously, the number of non-modelled days was used to determine the optimum window length (Canonaco et al., 2021). For a window length of 7 days, 4.2 % of the time points were not modelled. In all the other tested window lengths (14, 21 and 27 days), all data points were modelled. The PMF errors slightly decreased with longer window length, although
not significantly. While a shorter window is favorable since unique time periods (i.e., special pollution events) will be less propagated into the PMF results, the window should still be long enough to capture systematical changes and filter out short-term fluctuations. Therefore, the 14-day window length solution was chosen here.

The repeats per window are required for the study of the statistical uncertainties of the rolling PMF approach. On the one hand, the statistical uncertainty can be assessed by the application of the bootstrap technique, where the PMF input is randomly
resampled before each PMF initialization. If factors are constrained with *a priori* information (reference profiles or external time series), the rotational ambiguity has to be explored by a sensitivity analysis of the *a*-value. It has been shown by Canonaco et al. (2021) that the exploration of the solution space with the full *a*-value range (0 to 1) is not necessary unless high *a*-values were already required for the seasonal pretests. Furthermore, the random exploration of the possible *a*-value combinations (in contrast to explicitly checking every possible *a*-value combination) has proven sufficient. For this study, the *a*-values were
chosen randomly for each PMF repetition, as well as independently for each factor, ranging for 0 to an upper *a*-value of 0.4 for HOA, BBOA and CCOA ($\Delta a = 0.1$). The upper cut-off was determined based on the seasonal pretests as for solutions with higher *a*-values the POAs were subject to mixing of profiles.

8193 solutions (36.9 %) out of the total 22'200 single PMF runs generated during the rolling PMF approach were regarded as environmentally reasonable based on the criteria described above. All time points were modelled. Analysis of the scaled
residuals over time and variables (*m/z*) did not reveal any systematic errors, as shown in Fig. S4.

### 3.2.3    Rolling PMF results

The average factor profiles are presented in Fig. 2. The error bars show the standard deviation and represent the variability of the factor profiles over the full campaign. The first three factor profiles represent the constrained primary OA sources: HOA, BBOA and CCOA. They were constrained with an average *a*-value of 0.19, 0.21 and 0.18 for HOA, BBOA and CCOA,
respectively. The two OOA sources were separated by the fraction of *m/z* 44 ($f_{44}$) in their profiles. In general, the primary (constrained) source profiles showed less variation compared to the secondary (oxygenated) source profiles. Especially the





LO-OOA showed high variability in *m/z* 43 and *m/z* 44. Without the rolling PMF technique, this variability would be harder to be explored.

The average time series and the seasonal diurnal variations are presented in Fig. 3. HOA presented a morning and an evening

peak, consistent with the traffic rush hours during the respective season. The concentrations of BBOA and CCOA were much higher in the colder periods compared to summer. Furthermore, the emissions were more dominant during the night. Therefore, the two combustion OAs were mainly attributed to residential heating. Overnight, BBOA declined earlier than CCOA. We interpret this to increased fireplace activity during the late evening (in contrast to residential heating overnight with a greater share of coal). The diurnal profiles of MO-OOA and LO-OOA are different, especially during summer, where the LO-OOA

concentration decreases during the day due to dilution, evaporation and photochemical aging into MO-OOA. In general, the diurnals of the primary pollutants were strongly driven by vertical mixing with low mixing heights causing enhanced accumulation of pollutants during the night and strong mixing causing dilution of the primary pollutants from late morning to the sunset. More details on the influence of the planetary boundary layer height (PBLH) is not further discussed as measurements such as $^{222}$Rn concentrations (Zimnoch et al., 2014) were not available during this campaign.

The monthly average concentrations and contributions of each OA factor to the total OA mass is presented in Fig. 4. The fractional contribution of HOA to the total mass was most constant. A clear seasonal pattern could be observed for the solid fuel OA factors (BBOA and CCOA) as well as for the OOA factors (MO-OOA and LO-OOA). The contribution of MO-OOA and LO-OOA was generally higher during summer compared to winter. The seasonal dependence of BBOA and CCOA was associated with residential heating in Krakow, resulting in higher contributions in winter. During the colder periods, CCOA

was more dominant than BBOA, which is also consistent with previous offline measurements (Zimnoch et al., 2020). Comparing the two winter periods covered in this study, the contribution of the primary sources was higher in the first winter although it was a rather warm and clean winter. In contrast, the contribution of MO-OOA to the total OA mass was higher in the second winter compared to the first winter. The drop in primary BBOA and CCOA could be related to Krakow preparing households to switch from solid fuels to non-solid fuel (mostly gas) for residential heating as required by the resolution to ban

all solid fuel by September 2019 (Marshal's Office of the Malopolska Region, 2016). On the other hand, regional influences by contributions from aged BBOA and CCOA from the surrounding villages might be manifested in MO-OOA.

Coal combustion typically leads to co-emission of chloride as well as several heavy metals and metalloids such as As, Ga, Pb and Se (Rai et al., 2021). Xact data is available for the last month of the campaign. As expected, the correlation between the coal-related trace elements measured by the Xact and the OA factors (Fig. S5) was highest with CCOA (As ($R^2 = 0.63$), Cl ($R^2$

= 0.85), Ga ($R^2 = 0.49$) and Pb ($R^2 = 0.55$)). The correlation of CCOA and Se ($R^2 = 0.28$) was slightly lower than the correlation of LO-OOA and Se ($R^2 = 0.42$). This is probably related to the non-zero saturation vapor concentration of SeO$_2$ such that Se is probably more related to regional transport of coal combustion emissions. Se is not only a marker for coal combustion but can also be used to trace SO$_2$ oxidation in clouds and fog as it has a similar removal rate as sulfate (Chiou and Manuel, 1986).

Based on the diurnal variation of CCOA, we expect CCOA to be mostly from residential heating. However, some metal
components might also be emitted by industries rather than only coal combustion, which could explain the reduced correlation
with the CCOA factor for some elements.

While the eBC source apportionment works well for environments with only two combustion sources (i.e. traffic and wood
combustion emissions), it has not been proven robust for more than two combustion sources. While we expect the separation
of $eBC_{tr}$ to be relatively good, as α can be expected quite constant during the course of a year, the $eBC_{sf}$ separation is more
uncertain. Therefore, only the $eBC_{tr}$ was used to define reasonable HOA solutions. Seasonal multi-linear regressions were
performed to better understand the contribution of the two and three combustion sources (traffic and solid fuel combustion and
biomass burning and coal combustion) to the total eBC, respectively. However, the results cannot be interpreted conclusively.
Together with the relatively high OA to eBC ratios (Table S1), it can be assumed that there is at least one additional source
contributing to the total eBC concentration, e.g. industrial emissions. Further investigation through offline analysis or long-
term campaigns with parallel ACSM and Xact measurements and subsequent combined PMF could give more insight into
further emission sources in Krakow.

Overall, 36.9 % of all PMF runs met all of the acceptance criteria simultaneously and were accepted as environmentally
reasonable solutions. The rolling PMF in combination with the bootstrap resampling strategy and the random *a*-value approach
for the constrained factors, results in the repeated sampling of each time point $i$. The statistical and rotational uncertainty is
represented by the variability among the time points $i$. The uncertainty is described as the logarithmic probability density
function (pdf) of the standard deviation of each time point $i$ divided by the mean concentration of each time point $i$. As time
points with a low signal-to-noise ratio would pull the error calculations, the lognormal distribution was chosen to better
represent the PMF errors. As shown in Fig. 5, the relative PMF errors are ± 27.1 %, ± 26.1 %, ± 14.6 %, ± 21.8 % and ± 39.2 %
for HOA, BBOA, CCOA, MO-OOA and LO-OOA, respectively.

## 4    Conclusions

This work presents the first real-time long-term measurements of submicron aerosol particle composition in Krakow. The
concentration of NR-PM$_1$ follows a clear seasonal trend with the highest concentrations during winter and lowest
concentrations during summer. The strongest seasonal pattern is observed for chloride, where emissions are much higher in
winter due to coal combustion and the low temperatures favor partitioning of chloride to the particle phase.

The rolling PMF technique was successfully applied to long-term measurements in Krakow. Unlike traditional PMF, the rolling
approach allows time-dependent factor profiles. Five OA factors were identified during the whole measurement period: HOA,
BBOA, CCOA, MO-OOA and LO-OOA. Over the full campaign, the average HOA contribution to the total OA mass was
8.7 % and showed a distinct diurnal pattern with peaks during the traffic rush hours. BBOA and CCOA have a similar trend
with high contributions (up to 38 % and 55 %, respectively) during the winter months. The total OOA (MO-OOA + LO-OOA)



followed the opposite seasonal trend compared to the combustion OA sources. While MO-OOA contributed more during the winter, the contribution of LO-OOA was higher during summer. The rotational and statistical uncertainties were assessed using the bootstrap resampling strategy combined with the random $a$-value approach. The relative PMF errors were ± 27.1 %, ± 26.1 %, ± 14.6 %, ± 21.8 % and ± 39.2 % for HOA, BBOA, CCOA, MO-OOA, and LO-OOA, respectively.

Highly time-resolved, long-term measurements provide a basis for monitoring the impact of policies, such as Krakow's ban
of all solid fuels from September 2019 onwards, on the air quality. Further work is required to fully characterize the impact of European coal combustion on a regional and local level.

*Data availability.* The data presented in the text and figures as well as in the supplement will be available upon publication of the final manuscript (https://zenodo.org). Additional related data can be made available upon request.

*Competing interests.* F. Canonaco is employed by Datalystica Ltd., the official distributor of the SoFi Pro licenses. During the final revision of the manuscript, A. Tobler has also been with Datalystica Ltd.

*Author contributions.* AS, AKT, JN and KS were responsible for the instrumentation installation, data collection and/or instrument calibration. GM analyzed the aethalometer data. PR and MF were responsible for the Xact measurements. JB and MZ provided external data. AKT analyzed the ACSM data, performed the SA and wrote the manuscript. ASHP, JGS, JN, KR
and UB were involved with the supervision. FC, JGS, ASHP and UB assisted in the interpretation of the results. All co-authors contributed to the paper discussion and revision.

*Acknowledgment.* This work was financially supported by the EU Horizon 2020 Framework Programme via the ERA-PLANET project SMURBS (grant agreement no. 689443), the Swiss State Secretariat for Education, Research and Innovation (SERI; contract no. 15.0159-1), the COST action CA16109 Chemical On-Line cOmpoSition and Source Apportionment of
fine aerosoLs COLOSSAL grant and the related project Source apportionment using long-term Aerosol Mass Spectrometry and Aethalometer Measurements (SAMSAM, IZCOZ0_177063), and the Swiss National Science Foundation (starting grant BSSGI0_155846). Furthermore, this research was partially financed by the AGH UST grant 16.16.210.476 subsidy of the Ministry of Science and Higher Education. Alicja Skiba was partly supported by the EU Project POWR.03.02.00-00-I004/16.



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





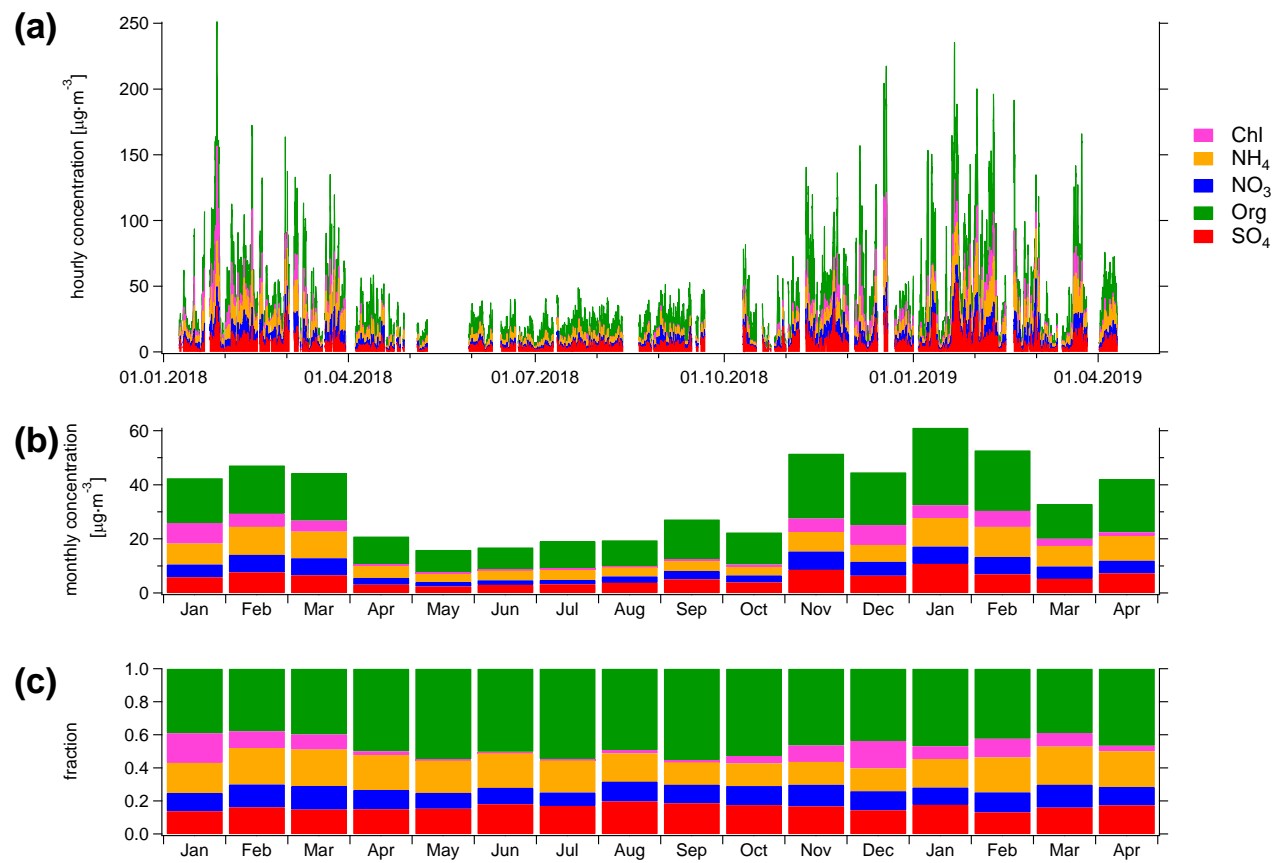

**Figure 1. (a) Hourly chemical composition of NR-PM1 in Krakow from 8 January 2018 to 10 April 2019. The bar plots show the monthly average chemical composition in (b) absolute mass loadings and (c) relative fractions. All times are in UTC.**






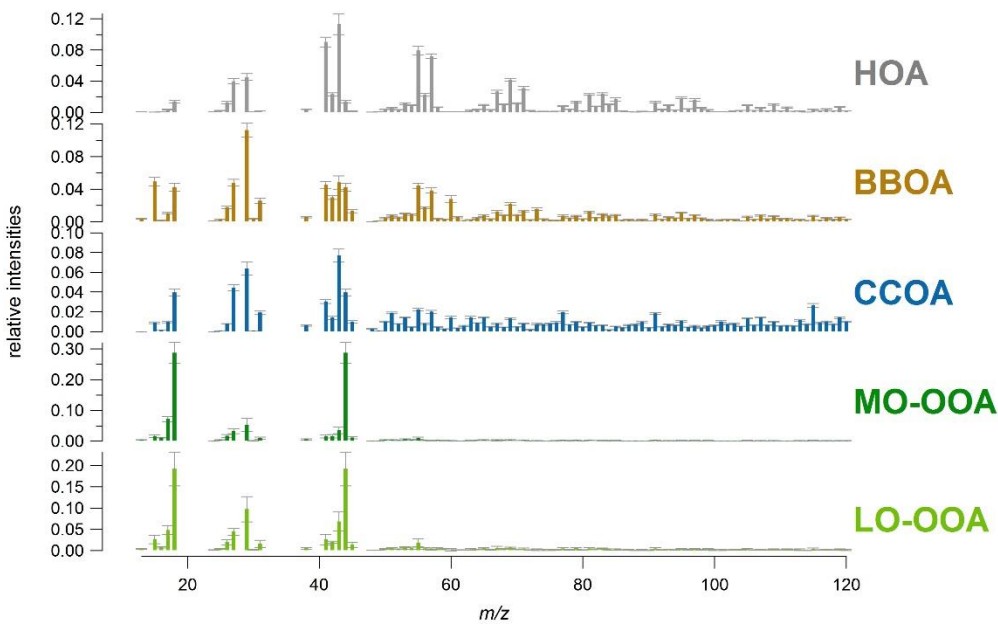

**Figure 2. Mean factor profiles of the five factors. The profile variability is represented by the grey error bars (standard deviation).**

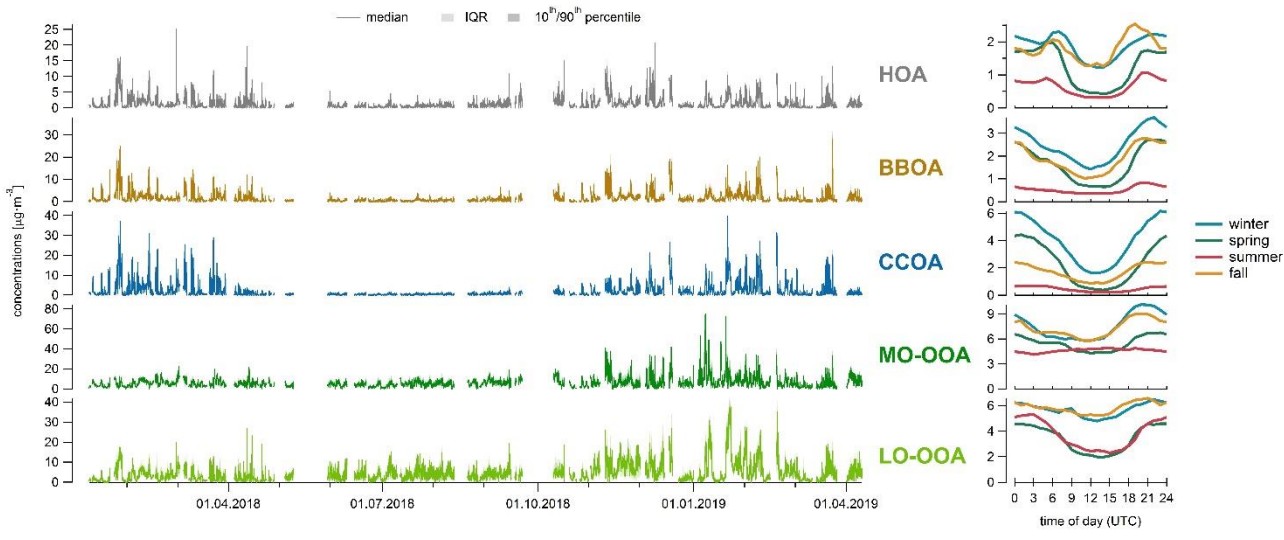


**Figure 3. Medians, interquartile ranges and 10th/90th percentiles of the time series (left) and mean values of the diurnal variations (right) of the five resolved PMF factors. The time is reported in UTC (1 and 2 hours behind local time during winter and summer, respectively).**





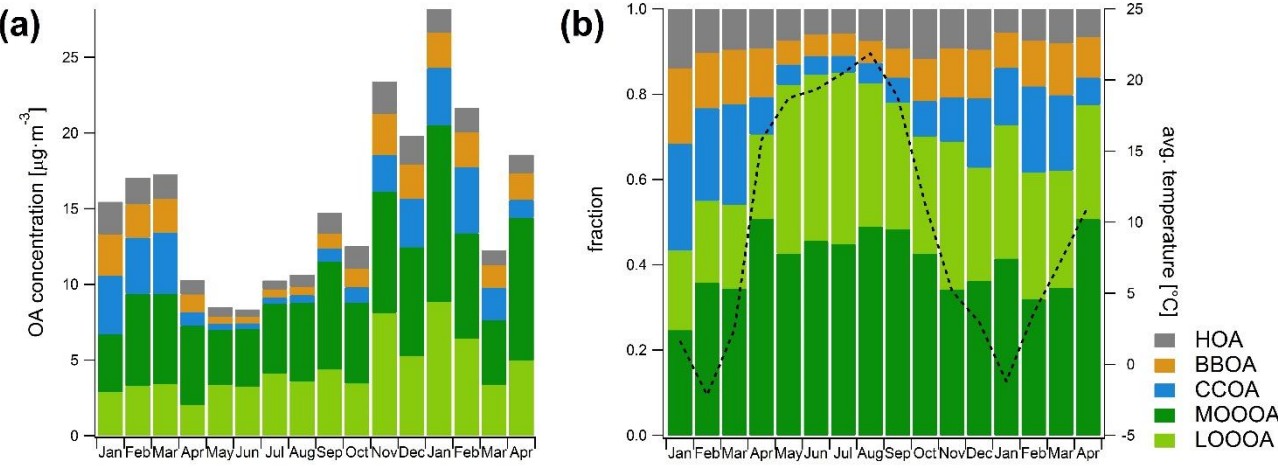


**Figure 4. (a)** Monthly concentrations and **(b)** contribution of the different sources to the total OA mass. While HOA has a similar contribution throughout the campaign, the solid fuel combustion related BBOA and CCOA show a clear seasonal pattern with highest contributions during winter. The contributions of MO-OOA and LO-OOA are highest during summer.






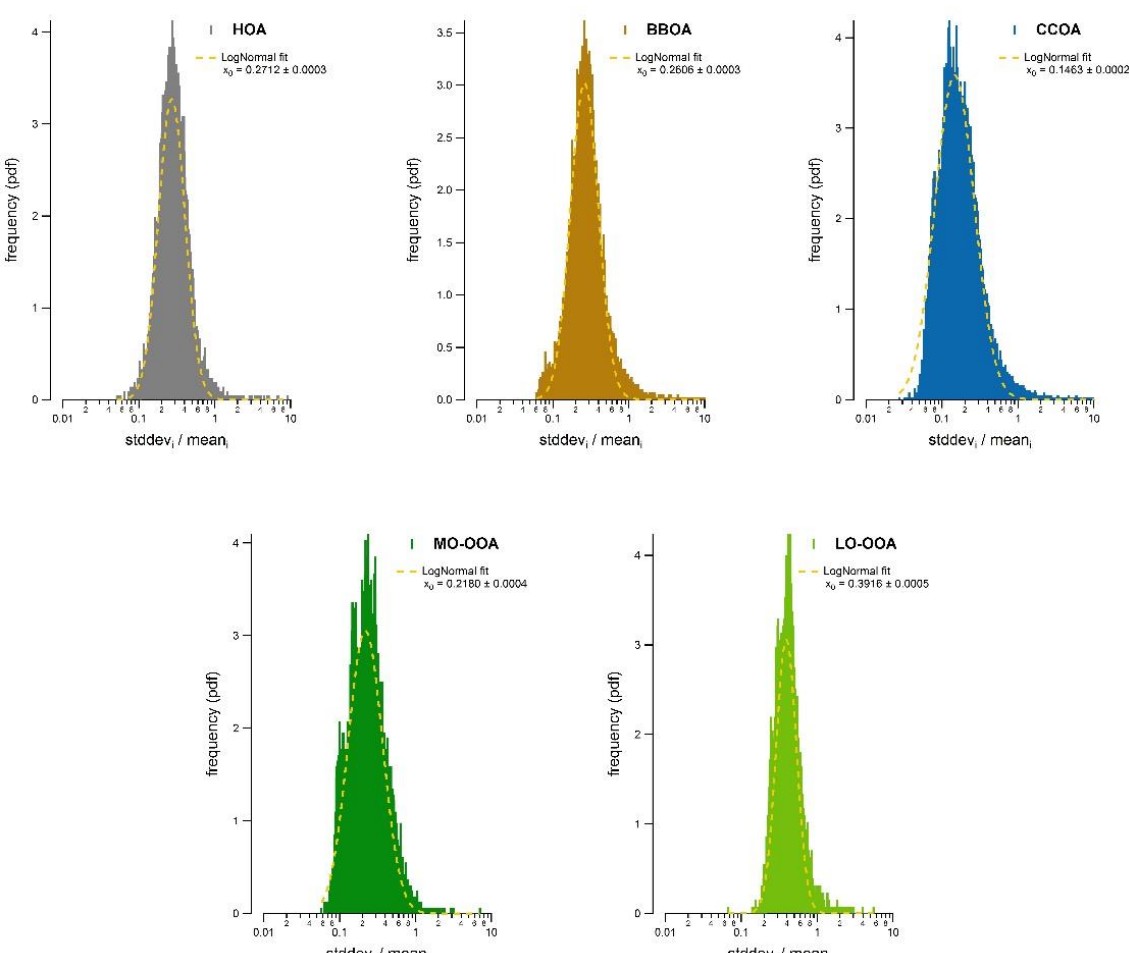

**Figure 5.** PMF error estimation of the five resolved PMF factors represented as logarithmic probability density functions (pdf) of the standard deviations of each time point $i$ divided by the mean concentration of each time point $i$.