# Peer review of "Characterization of NR-PM1 and source apportionment of organic aerosol in Krakow, Poland"

_Atmospheric Chemistry and Physics, 2021_

## Author Response (AR1)

**Characterization of NR-PM1 and source apportionment of organic aerosol in Krakow, Poland**

Anna K. Tobler[1,2], Alicja Skiba[3], Francesco Canonaco[2], Griša Močnik[4,5], Pragati Rai[1], Gang Chen[1], Jakub Bartyzel[3], Miroslaw Zimnoch[3], Katarzyna Styszko[6], Jaroslaw Nęcki[3], Markus Furger[1], Kazimierz Różański[3], Urs Baltensperger[1], Jay G. Slowik[1] and André S. H. Prévôt[1]

[1]Laboratory of Atmospheric Chemistry, Paul Scherrer Institute, 5232 Villigen PSI, Switzerland
[2]Datalystica Ltd., Park innovAARE, 5234 Villigen, Switzerland
[3]AGH University of Science and Technology, Faculty of Physics and Applied Computer Science, Department of Applied Nuclear Physics, Krakow, Poland
[4]Condensed Matter Physics Department, J. Stefan Institute, Ljubljana, Slovenia
[5]Center for Atmospheric Research, University of Nova Gorica, Ajdovščina, Slovenia
[6]AGH University of Science and Technology, Faculty of Energy and Fuels, Department of Coal Chemistry and Environmental Sciences, Krakow, Poland

*Correspondence to*: André S.H. Prévôt (andre.prevot@psi.ch)

**Author's response to Anonymous Referee #1**

We thank anonymous Referee #1 for the careful revision and useful comments to improve the manuscript. The referees' original comments (in *italic typeset*) is followed by the author's answer (in regular typeset). Changes to the manuscript are indicated in green font.

*This paper presents an analysis of a Q-ACSM dataset from Krakow, applying the rolling window ME2 algorithm. The methods are appropriate and the paper is overall well written, but the level of analysis is very shallow and does not extend beyond the validation of an appropriate solution set from the analysis. Given no scientific conclusions are presented, this to me seems to be a very clear-cut case of a manuscript that should be published as a measurement report rather than a research article. If the authors wish to present this as a technical development of the apportionment tools, then this could be classified as a technical note. However, for this to be a valid technical note, much more detail should be presented on the methodology and how this advances the current state of the art.*

> We strongly believe that our manuscript fits well into this journal as a research article. We understand that the structure of the manuscript would also fit into a measurement report and many technical details were discussed in the result section, as referee #2 also pointed out. However, we believe this manuscript goes far beyond the validation of a new analysis approach (that has only been demonstrated for two other locations, both in Switzerland, so far). To highlight this, we added additional discussion about the seasonal trends of OA and an approach to explain the OOA source origins. Furthermore, we can conclude from our findings that the contribution of coal combustion to organic aerosol is substantial and mostly related to residential burning because of the pronounced yearly cycle. Other coal

related sources of OA are likely not as important. This important finding is also relevant to other coal burning regions in Europe and Asia and has not been highlighted enough in the manuscript, therefore we added:

Line 31: Not only BBOA but also CCOA is associated with residential heating because of the pronounced yearly cycle where the highest contributions are observed during wintertime.

Line 304: Due to the pronounced yearly cycle (similar to BBOA) other coal related sources of OA are likely not as important. It can be assumed that also in other regions in Europe and Asia, where residential coal combustion is still practiced and high OA mass loadings are measured (like in the Western Balkans or Northern India), residential coal combustion is a substantial emission source.

Line 377: Residential heating is the dominant source for both BBOA and CCOA.

*If the paper is reclassified as a measurement report (my preferred option), then I can recommend publication subject to the following comments:*

1. *The rejected 6 factor solution should be shown in the supplement.*

   The 6 factor seasonal solution was added to the supplement. Based on the mass spectral profiles either the OOAs (spring, summer, fall) or CCOA/OOA (winter) was mathematically split when increasing the number of factors.

   A reference to the figure in the supplement was made in the manuscript:

   Line 217: Increasing the number of factors to 6 or more factors led to splitting of either the OOA or the OOA/CCOA factors (Fig. S5).

[Figure]

**Figure S1. Rejected 6-factor solution based on seasonal PMF, due to splitting of factors.**

2. *Describing how eBC$_{tr}$ and metal were used to constrain and validate some of the factors is a little unsatisfactory. The authors should present correlations visually, e.g. with scatter plots. The same should be done with eBC$_{wb}$, if only to demonstrate the weak performance*

*of this. Related to this point, the authors should be consistent between whether they call the Aethalometer data product 'eBC$_{wb}$' or 'eBC$_{sf}$'.*

We added the scatter plot of eBC$_{tr}$ and HOA as well as the scatter plot of eBC$_{sf}$ and (BBOA+CCOA) to the supplement and a short statement to the manuscript:

Line 348: In the average solution, HOA and eBC$_{tr}$ show a correlation of $R^2 = 0.73$, while the correlation of eBC$_{sf}$ and the sum of BBOA and CCOA is $R^2 = 0.88$ (Fig. S8). The two different slopes for the two winters in the eBC$_{sf}$ versus (BBOA + CCOA) plot could be related to a change in the solid fuel composition used for heating as a preparation for the solid fuel ban.

[Figure]

**Figure S2. (a) eBC$_{tr}$ versus HOA and (b) eBC$_{sf}$ vs (BBOA + CCOA).**

While the terminology eBC$_{tr}$ and eBC$_{wb}$ is used to refer to previous studies in Alpine valleys where only two combustion sources were present, we use eBC$_{tr}$ and eBC$_{sf}$ for the source apportionment results in Krakow, where we expect at least three combustion sources. With the Aethalometer model, we cannot separate biomass burning from coal combustion and therefore we use eBC$_{sf}$. For clarification we added a small explanation to the manuscript:

Line 124: Sandradewi et al. (2008) presented a model to separate biomass burning (eBC$_{wb}$) and traffic (eBC$_{tr}$) contributions in environments dominated by those two combustion types.

Line 131: In Krakow, we expect at least three combustion sources that contribute to eBC, namely traffic, biomass burning and coal combustion. We expect to separate traffic emissions from solid fuel (biomass burning and coal combustion) emissions, therefore the terminology eBC$_{tr}$ and eBC$_{sf}$ will be used in this manuscript, when referring to eBC source apportionment in Krakow.

3. *It's really not apparent what criteria were used to determine the AAE values in figure S2. Do these correspond to a particular level in the PDF?*

We had added the following explanation to the Supplement:

Source apportionment uses source specific values of the Ångström exponent (AAE). The traffic features the AAE value between 0.9 and 1.1. The value for solid fuel is less well determined as it depends on the efficiency of combustion. These source specific values are supposed to be representative for the source, but are in fact a single value representing the center of a distribution for this particular source. In the absence of validation measurements (for example C14, (Zotter et al., 2017)), plotting the probability density function can serve as a guide for the determination of these values as seen in Fig. S3. Source specific values for traffic $AAE_{tr}$ = 0.85 and solid fuel $AAE_{sf}$= 1.9 were selected. We see in Figure S3 the AAE probability density function (PDF) for all Krakow Aethalometer absorption data calculated in two different ways. First, the AAE was calculated as the ratio of the logarithms of the absorption coefficient:

$$AAE = \frac{\ln(b_{370}/b_{950})}{\ln(950/370)}$$

and it is shown in blue in Figure 3. The AAE obtained from the fit of the absorption coefficient as a function of the wavelength (from 370 nm to 950 nm) is shown in green with applying a very stringent filter $r^2$>0.99 for the fit. The resulting PDF (in green) substantially shrinks the tails of the PDF compared to the PDF of the AAE as a ratio. This filtering allows only the "best" fits and the end PDF values are the ones that we can ascribe to two sources - the tails for this stringent-filtered data end at the values that we chose for the source specific AAE values.

[Figure]

**Figure S3. The absorption Ångström exponent (AAE) probability density function: AAE calculated from the ratio of the 370 nm and 950 nm channels (blue) and from the fit off all wavelengths from 370 nm to 950 nm and filtered for fit $r^2$>0.99.**

4. *Using the 'Chl' product to constrain the CCOA factor could be problematic because ammonium chloride is semi volatile and may vary with temperature and relative humidity. Furthermore, and abundance of nitric or sulphuric acid may displace it from the particles, which would modulate the data in ways not representative of the actual coal OA.*

*Specifically, this could put an artificial diurnal cycle on the factor. Can the authors verify this is not the case? Without evidence to the contrary, I would expect it safer not to use this as a constraint.*

We agree with the referee that using Chl to constrain CCOA is not ideal and could be problematic if it is used as a criterion during the whole campaign period because of the semi volatile nature of chloride. To minimize the effect of volatility driven partitioning of chloride, we use this constraint only during the cold period and exclude this criterion from 31 March to 2 November 2018. With using the chloride criterion only during the cold period and the fact that chloride and CCOA have the same diurnal pattern in the seasonal (winter) solution, we are confident that this is a valid criterion.

5. *The CCOA factor contains a lot of small signals at high m/z channels that do not display much of a mass spectral pattern. Can the authors be sure that this is 'real' signal and not noise?*

Small signals do not necessarily mean that there is not real data. We used the signal-to-noise ratio (*S/N*) to determine whether a signal (*m/z*) is noisy or not. If *S/N* is smaller than 2, the specific *m/z* is regarded as a weak variable and is downweighted by a factor of two, if *S/N* is smaller than 0.2, the specific *m/z* is regarded as a bad variable and is downweighted by a factor of ten. In our case, all signals at high *m/z* show a mean *S/N* larger than 2 and are not downweighted (see Figure below).

[Figure]

6. *The percentiles on figure 6 are not visible to me.*

There is no Fig. 6 in the original manuscript. We assume the referee is referring to Fig. 3. We agree with the referee. The length of the campaign and the relatively small percentiles make it hard to adequately represent the results. This highlights the good performance of the rolling PMF. The PMF errors are shown in Fig. S7.

7. *Were correlations between any of the metals and any of the other factors noted?*

The correlation of four out of the five elements that are typically related to coal combustion and the OA factors are clearly highest with the CCOA factors. For example for CCOA and Cl $R^2$ = 0.85, while the next highest correlation only shows $R^2$ = 0.48 (HOA and Cl). The only exception is the correlation with Se which is discussed in the manuscript. To clarify this in the manuscript we added:

Line 339: The correlation with the other factors was clearly lower ($R^2$ = 0.39 for As and LO-OOA and HOA, $R^2$ = 0.48 for Cl and HOA, $R^2$ = 0.26 for Ga and LO-OOA and $R^2$ = 0.41 for Pb and HOA).

A comprehensive analysis of the Xact results including a comparison with the OA factors is subject to a future publication.

**Author's response to Anonymous Referee #2**

We thank anonymous Referee #2 for the careful revision and useful comments to improve the quality of the manuscript. The referees' original comments (in *italic typeset*) is followed by the author's answer (in regular typeset). Changes to the manuscript are indicated in green font.

*This article aims at apportioning the main constituents of submicron organic aerosols in Krakow, Poland, using state-of-the-art monitoring methodologies, from winter 2018 to spring 2019, i.e., before the ban of solid fuel combustion within the city is coming into effect. The manuscript can represent a substantial contribution to scientific progress in the fields of air quality and atmospheric chemistry, notably bringing new method and data. It is very well written and rather well structured, though maybe slightly too much concise. I would recommend its publication in ACP after the following major revisions:*

1. *Quantification issues (i.e., concentration absolute values) could be of further particular interest when it will come to evaluate the impact of the ban of solid fuel on OA ambient air concentrations. In this view, the quality control of ACSM measurements could be reinforced here by mean of comparisons with independent co-located measurements, mainly with available PM2.5 monitoring data (which shall be added into Figure 1a and 1b). RF and RIE values retrieved from regular calibrations could also be presented and discussed as supporting material. Similarly, it should be stated more clearly than eBC concentrations are (apparently) used as provided by the AE33 instrument, i.e., no further application of*

*any correction factor. This is possibly leading to some kind of 'underestimation' of the OA/eBC ratios presented in Table S1, when compared to other studies, e.g., in Europe.*

Unfortunately, we do not have any co-located total $PM_{2.5}$ measurements, however, we have reference data available from the monitoring station ("Bujaka station") run by the Chief Inspectorate for Environmental Protection, located ca. 6.8 km south-east of AGH monitoring station.

Although, we are comparing here $NR-PM_1$ and total $PM_{2.5}$, where the stations are 6.8 km apart and in a urban area, which means that local sources could be contributing quite differently, there is still a decent correlation of $R^2 = 0.67$ and a reasonable slope of 0.77.

While we forego to add it to Fig. 1, because the stations are still 6.8 km apart, the graph was added to the supplementary together with a short discussion of the calibrations:

Line 257: Regular calibrations ensured the quantification and the comparison of $NR-PM_1$ with total $PM_{2.5}$ from the Bujaka station further support the reported concentrations (Sect. S1).

In the supplement (new Section S1): The ACSM was routinely calibrated with $NH_4NO_3$ and $(NH_4)_2SO_4$ to determine the $RF_{NO3}$, $RIE_{NH4}$ and $RIE_{SO4}$ in full scan mode, meaning that the same scanning protocol was used during the calibration as during ambient measurements (Freney et al., 2019). In addition, the ACSM was also calibrated with $NH_4Cl$ to determine the RIE_Chl', following the procedure described by Tobler et al. (2020), where the RIE_Chl' is only based on the ion signals of frag_HCl and does not include frag_Cl. An average $RF_{NO3} = 4.68 \pm 1.66 \times 10^{-11}$ A($\mu$g m-3)-1 was applied together with an $RIE_{NH4} = 2.43 \pm 0.58$, $RIE_{SO4} = 0.38 \pm 0.11$ RIE_Chl' $=0.41 \pm 0.17$.

Unfortunately, co-located total PM2.5 measurements were not available at AGH University for quality control. However, total PM2.5 reference data was available from the monitoring station ("Bujaka station") run by the Chief Inspectorate for Environmental Protection, located ca. 6.8 km south-east of the AGH monitoring station. Taking into account that the two stations are 6.8 km apart and in a urban area, which means that local sources could be contributing quite differently, there is still a decent correlation of $R^2 = 0.67$ and a reasonable slope of 0.77.

[Figure]

**Figure S4. NR-PM₁ measured by the ACSM at AGH University versus total PM₂.₅ measured at the Bujaka station.**

The Aethalometer AE33 does not need any post-processing for loading effects – the measurement features collection of the sample on the filter in two sample spots and then compensates for the loading by extrapolating to the fresh filter (Drinovec et al., 2015). We had checked this by plotting the dependence of BC on the attenuation (ATN). BC should be linear with the rate of change of ATN and should not depend on the ATN value – the BC(ATN) slope should be zero, which can be seen in the figure below: BC(ATN) for the AE33 wavelength 470 nm, for the period Jan 2018 to Apr 2019. The OA/eBC ratios are therefore unaffected by the measurement method.

[Figure]

2. *Authors are nicely explaining Line 247-250 that: "Although the aethalometer model (Sandradewi et al., 2008) was developed for environments dominated only by two combustion sources, namely traffic and biomass burning, the model still works well enough to separate liquid (traffic) and solid fuels (biomass burning and coal combustion)". One can further argue that liquid fuel combustion is not necessarily related to road transport only. For these reasons, eBCtr and eBCwb could be replaced by eBClf (for liquid fuel) and eBCsf (for solid fuel) throughout the manuscript. This might probably be mimicked then in many further studies using the so-called Aethalometer model.*

We agree with the referee that this should be looked into in more detail. However, we do not feel comfortable to propose in this paper a new nomenclature that still would be debatable. Also with liquid fuels in case of very inefficient burning, one can produce aerosol with high absorption exponents. Therefore, the definition may need to be on the axis of efficiency, which is often related to the fuels but not necessarily. So we would like to stick to the terminology used so far but plan to submit a stand-alone note on this in the future.

3. *To avoid this paper being seen mainly as a technical (or methodological) paper, sub-sections 3.2.1 and 3.2.2. (as well as Figure 5) should rather be discussed within section 2 and/or in supporting material. Moreover, to go more clearly beyond a measurement report, the seasonal origins of OOA sub-fractions could be investigated and discussed in more details.*

Thank you for this great suggestion, we fully agree here with the referee and changed the manuscript accordingly. Section 3.2.1 is now discussed within Sect. 2, whereas Sect. 3.2.2 as well Fig. 5 and the corresponding error calculation explanation found in Sect. 3.2 can be found in the supplement.

We agree that the seasonality of the OA factors, in particular the OOA factors could be discussed in more details. Therefore, we added to the manuscript:

Line 314: The contribution of the five OA factors as a function of OA mass loadings during the seasons is presented in Fig. 5. In spring, the low OA mass loading is dominated by the two OOA fractions, in particular by MO-OOA. With increasing OA mass loadings and decreasing temperature in spring, the POA contribution changes drastically. The contribution of CCOA increases gradually from 7 % to 30 % and the contribution of BBOA increases from 7 % to 18 %, while the MO-OOA contribution decreases from 55 % to 21 % and the LO-OOA contribution decreases only from 25 % to 19 %. In summer, a small increase of BBOA (4 % to 11 %) can be observed, while CCOA stays constant with a contribution of 4 %. The increase in BBOA could be related to an increase in outdoor activities related to the warmer weather. In fall, a continuous increase of BBOA and CCOA (5 % to 16 % and 8 % to 12 %, respectively) and decrease of LO-OOA (from 37 % to 19 %). with increasing OA mass loadings is observed, while MO-OOA has a rather constant contribution of 40 %. In winter, BBOA contributes around 10 % to the total OA mass over all OA mass loadings, while the contribution of CCOA continuously increases from 10 % to 23 % with increasing OA mass loadings. While the LO-OOA contribution slightly increases (from 23 % to 26 %), the MO-OOA contribution decreases from 49 % to 30 %. The mass fraction of HOA during all seasons and OA mass loadings is rather constant. In general, low OA mass loadings are dominated by the OOA factors while the POA factors

gain importance with increasing OA mass loadings. The highest seasonal contribution of POA is observed in spring, however, at the temperature similar to in winter. These results emphasize the major role of primary sources during pollution events, in particular POA resulting from solid fuel combustion. Furthermore, in spring, summer and winter, LO-OOA is favored compared to MO-OOA in periods with high OA mass loadings.

[Figure]

**Figure 5. Seasonal contributions of the OA sources (HOA, BBOA, CCOA, MO-OOA and LO-OOA) as a function of the total submicron OA mass loadings. The average bin temperature is shown in red.**

Figure 6 shows the temperature dependence of the OOA factors. In summer, the OOA concentrations substantially increase with temperature as expected for the formation of biogenic SOA (Daellenbach et al., 2017). In winter, high OOA concentrations are observed with lower temperature. This could be related to SOA formation from residential heating precursors. In spring and fall, no clear trend is observed. The high concentration at lower temperature is linked to the cold period in spring, when also high POA concentrations from CCOA can be observed.

[Figure]

**Figure 6. Daily average concentrations of the OOA factors versus the daily average temperature for summer, winter and spring/fall.**

*Other comments:*

*- Lines 52-53 are mixing type of fuel and type of activities. Does "coal" means heating here? and/or what type of emissions and (combustion?) processes are linked to "steel, cement and metal industries" mentioned here?*

The explicit emission processes of the industrial sources were not explored in these studies. Most likely, these are probably linked directly to products handled by the industry, and therefore the high contribution of these elements to this factor. Also, the studies did not focus on the main contribution to the coal factor. It can be attributed to residential heating as well as other coal combustion, depending on the study and which elements are attributed to this factor.

*- Line 81-82: not clear why this change of set-up (AE33 upstream and then downstream the ACSM sampling line, including dryer) was achieved, and if this might have any influence on eBC quantification (?).*

The set-up was changed, to avoid condensation in the AE33 during the warmer summer months, which we did not think of initially. Overall, the AE33 data measured before the dryer is noisier, but since it is 1 min resolution this should only have a minor effect on the 30-min average data.

*- Line 100: can be changed into: "... the method proposed by Middlebrook et al. ... ". And, line 101, not clear why CE of 0,5 instead of 0,45, as also proposed by Middlebrook et al.*

Indeed, Middlebrook et al. suggest a CE of 0.45 with focus on the composition dependent CE (CDCE). It terms of what the best base CE values is, only a small number of data sets was included in that study. The CE is typically determined by mass closure measurements against other instruments and rarely single particle light scattering measurements and due to e.g. lens transmission or particle size distribution, there is an instrument-to-instrument uncertainty with this number. Therefore, the manufacturer's default of 0.5 was regarded as more reliable.

*- Line 194: "... this variation is expected [or assumed, or ...] to be associated ..."*

The sentence was changed accordingly.

*- Line 233: "Since the coal profiles may strongly depend on the type of coal". This is also true for biomass burning aerosols. Sensitivity tests could be conducted using other BBOA profiles than the one proposed by Ng et al. for US.*

We fully agree with the referee that also the BBOA spectral profile strongly depends on the type of biomass, the combustion conditions/efficiency and the ambient temperature (i.e. partitioning). From this perspective, one could argue that the BBOA profile by Ng et al. from the US might fit worse than e.g. the BBOA profile from Crippa et al. from Paris as the studies in the US typically involve more wildfires whereas the reference profiles from Europe include more domestic heating.

With this particulate dataset, we were not able to separate a clean BBOA (and CCOA) factor without constraints. Therefore, we constrained our data with different BBOA reference profiles and compared the resulting CCOA factors. Empirically, we found that when using the BBOA profile from Ng et al. we get better results for our CCOA factor. By no means are we arguing that the BBOA profile by Ng et al. is the best profile to use in other analyses but we would like to highlight the importance of constraining PMF with different reference

profiles. In this study, the reference profile was chosen based on a model performance metric, rather than the similarities of the fuel type, burning conditions and chemical processes.

After deciding on using the reference profile of Ng et al. for the constraints, we also ran sensitivity tests for BBOA (only for the winter seasons). This was done by loosely and randomly constraining the BBOA profile and running a bootstrap analysis and all the runs with a reasonable CCOA profile were kept. The BBOA used to constrain the rolling PMF is the bootstrapped result of these tests.

*- Line 348-349: not quite clear why relatively high OA/eBC ratios are possibly revealing other eBC sources, nor what type of eBC emissions from industrial activities are suggested here (liquid or solid fuel burning?).*

There was actually a mistake in the manuscript. The data shows low (not high) OA/eBC ratios or high eBC/OA, respectively. We changed the manuscript accordingly:

Line 351: Together with the relatively low OA to eBC ratios (Table S1), it can be assumed that there is at least one additional source contributing to the total eBC concentration, e.g. industrial emissions or plastic combustion.

*- Scatterplot between eBCsf and the sum of BBOA + CCOA may be presented and further discussed.*

A similar suggestion was brought up by referee #1 (point 2). The scatter plots were added to the supplement and briefly discussed in the manuscript. For details, please refer to point 2 raised by referee #1.

*- Line 350: a deeper investigation of combined PMF analysis mixing ACSM+AE33+X'act datasets already available for the period March-April 2019 might also bring valuable information for this purpose?*

We agree that the combination of all the datasets could bring valuable insight. However, a thorough analysis should be done carefully. Therefore, a deeper investigation of the overlapping period of all three instruments (ACSM, Xact and AE33) is subject to future work and publication.

**References**

Drinovec, L., Mocnik, G., Zotter, P., Prevot, A. S. H., Ruckstuhl, C., Coz, E., Rupakheti, M., Sciare, J., Muller, T., Wiedensohler, A., and Hansen, A. D. A.: The "dual-spot" Aethalometer: an

improved measurement of aerosol black carbon with real-time loading compensation, Atmos. Meas. Tech., 8, 1965-1979, https://doi.org/10.5194/amt-8-1965-2015, 2015.

Zotter, P., Herich, H., Gysel, M., El-Haddad, I., Zhang, Y. L., Mocnik, G., Huglin, C., Baltensperger, U., Szidat, S., and Prevot, A. H.: Evaluation of the absorption Angstrom exponents for traffic and wood burning in the Aethalometer-based source apportionment using radiocarbon measurements of ambient aerosol, Atmos. Chem. Phys., 17, 4229-4249, https://doi.org/10.5194/acp-17-4229-2017, 2017.